# G Protein-Coupled Receptors and the Rise of Type 2 Diabetes in Children

**DOI:** 10.3390/biomedicines11061576

**Published:** 2023-05-29

**Authors:** Alessia Dallatana, Linda Cremonesi, Maddalena Trombetta, Giulio Fracasso, Riccardo Nocini, Luca Giacomello, Giulio Innamorati

**Affiliations:** 1Department of Surgical Sciences, Dentistry, Gynecology and Pediatrics, University of Verona, 37134 Verona, Italy; alessia.dallatana@univr.it (A.D.); linda.cremonesi@univr.it (L.C.); riccardo.nocini@univr.it (R.N.); luca.giacomello@univr.it (L.G.); 2Section of Endocrinology, Diabetes and Metabolism, Department of Medicine, University of Verona, 37124 Verona, Italy; maddalena.trombetta@univr.it; 3Department of Biomedical Sciences, University of Padova, 35131 Padova, Italy; giulio.fracasso@unipd.it

**Keywords:** type 2 diabetes, diabetes in youth, GPCRs, β cells, glucose homeostasis

## Abstract

The human genome counts hundreds of GPCRs specialized to sense thousands of different extracellular cues, including light, odorants and nutrients in addition to hormones. Primordial GPCRs were likely glucose transporters that became sensors to monitor the abundance of nutrients and direct the cell to switch from aerobic metabolism to fermentation. Human β cells express multiple GPCRs that contribute to regulate glucose homeostasis, cooperating with many others expressed by a variety of cell types and tissues. These GPCRs are intensely studied as pharmacological targets to treat type 2 diabetes in adults. The dramatic rise of type 2 diabetes incidence in pediatric age is likely correlated to the rapidly evolving lifestyle of children and adolescents of the new century. Current pharmacological treatments are based on therapies designed for adults, while youth and puberty are characterized by a different hormonal balance related to glucose metabolism. This review focuses on GPCRs functional traits that are relevant for β cells function, with an emphasis on aspects that could help to differentiate new treatments specifically addressed to young type 2 diabetes patients.

## 1. Type 2 Diabetes Prevalence on the Rise in Youth

Type 2 diabetes is a metabolic disease characterized by chronic hyperglycemia resulting from a progressive loss of β cell insulin secretion, frequently with the background of insulin resistance [1]. The incidence of type 2 diabetes in youth has increased worldwide and it is linked to rising levels of obesity, physical inactivity and poor diet. The higher prevalence of type 2 diabetes in Native Americans, Pacific Islanders, Hispanics, African Americans and South East Asians [2] has been attributed to a complex interaction between genetic and environmental factors. In the United States, approximately 3700 youths are diagnosed with type 2 diabetes every year and the number is expected to quadruple from 2010 to 2050 [3,4]. The number of children and adolescents living with type 2 diabetes has been increasing over the last decade in Europe as well [5]. Type 2 diabetes in this age group is becoming a global public health issue.

Modifiable risk factors such as poor diet and sedentary life are clearly achievable targets; however, a better knowledge of the pathophysiology of the disease could reveal additional strategies to treat or prevent young onset of type 2 diabetes. In respect to adult type 2 diabetes, youths show a lower response rate to lifestyle and metformin therapy and a relatively higher aggressiveness of the disease with earlier occurrence of complications [2,6,7].

During puberty, transient insulin resistance is physiological. The organism reacts with a compensatory increase in basal and stimulated insulin secretion. Normal responsiveness to insulin is usually recovered by adulthood [8] but may not resolve in obese adolescents that face higher glycemic challenges. Prolonged hyperglycemia causes a constant stimulation of β cells which over time lose functionality and die due to oxidative stress induced by glucotoxicity and lipotoxicity.

Pubertal development is also associated with several hormonal changes that involve GPCRs and can directly influence insulin secretion. In fact, the plasma membrane of β cells is populated by GPCRs such as the growth hormone (GH) receptor [9], the G protein-coupled estrogen receptor 1 (GPER1) [10] and the G protein-coupled receptor family C member A (GPRC6A) [11] that sense estrogen and osteocalcin, respectively. Fetal and neonatal β cells actively proliferate during pancreas development and during adolescence. The process declines and virtually ends in adulthood. Whether under certain condition β cells can start to expand again remains controversial [12]. Until puberty, GH promotes β cell proliferation. Recent work, in human and mouse β cells, demonstrated that the effect is mediated by autocrine/paracrine serotonin (5-hydroxytryptamine [5-HT]) activation of the GPCR HTR2B [13]. Exendin-4 (Ex-4), an agonist of the glucagon-like peptide 1 receptor (GLP-1R) was reported to stimulate human β cell proliferation in juvenile but not adult islets [14]. Other GPCR agonists were proven to induce β cell proliferation in mice. The same stimuli failed to effectively replicate in humans [15], possibly because in mice β cell proliferation rate is generally higher and persists in adult animals [12]. If trials failed in adults because of the lack of β cell proliferation rate, it would be desirable to develop new trials directed towards younger courts of patients, dedicating particular attention to centering the correct age window.

Obesity could contribute to the etiology of type 2 diabetes also by counter-balancing the anti-inflammatory effect of sex hormones [16,17,18]. Puberty is characterized by increased anti-inflammatory and antioxidant mechanisms [19] and an imbalance of glucose metabolism while the development of several organs and endocrine communication axes is still in progress could partially contribute to β cell dysfunction due to inflammatory stress.

The first-line treatment for type 2 diabetes in adults is metformin but the TODAY study reported that this monotherapy fails in 50% of pediatric patients within the first year of administration [20]. Other treatments for young patients include insulin and three recently approved agonists of the glucagon like peptide 1 (GLP1), liraglutide, exenatide and semaglutide, which stimulate insulin secretion and appear to be the most cost-effective strategy [21]. Another anorectic strategy to treat obese adults became also available for pediatric type 2 diabetes patients with the approval of Phenteramine/Topiramate. This treatment acts in the central nervous system (CNS) by inhibiting norepinephrine uptake and reducing glutamate and neuropeptide Y levels [22]. Therapies aimed at treating glycemia disorders are now shifting towards upstream targets, and indirectly regulate insulin release while preserving the concerted dynamics of incretins in response to nutrients uptake from the intestine. It will be crucial to tailor these new therapies with consideration of the aforementioned specific developmental dynamics of pediatric patients.

GPCRs are one, if not the most, successful class of drug targets. An enormous number of molecules have been developed to control their signaling. Specific agonists could likely be exploited to promote β cell proliferation or antagonize signaling that supports quiescence [23] and more in general to preserve the harmonization of endocrine communications required to control glucose metabolism during childhood [24]. Being able to intelligently intervene on GPCR signaling to compensate distortions due to sedentariness and hypercaloric diet could be key to prevent the damages related to the so-called modern “obesogenic environment” for which we did not evolve [25].

## 2. Learning from Yeast History

Glucose is the preferred carbon and energy source from prokaryotes to metazoans. It has been hypothesized that prokaryotes specialized transporters delegated to glucose uptake to become sensors and report the external abundance of glucose. Eukaryotes’ glucose metabolism relies on mitochondria and chloroplasts evolved from endosymbiotic organisms. The fusion operation implied a major genomic redistribution of genetic information toward the nucleus [26]. Glucose sensors were likely imported with mitochondria to allow eukaryote cells to regulate transcription and adapt the enzymatic repertoire to the most appropriate metabolism [27]. The evolution of this successful molecular machinery led to the largest class of sensors in nature. In humans, hundreds of GPCRs populate the plasma membrane and intracellular organelles, including mitochondria. This signaling machinery is fundamental to the physiology of any cell in adult life, but also to driving development. For example, the key hormone for the regulation of blood insulin levels, i.e., GLP1, is known to regulate early embryonic development of the whole pancreas and the differentiation of β cells through its action on the transcription factor PDX-1 [28]. Other important functions involved in glucose homeostasis also rely on GPCRs: insulin secretion is strongly regulated through glucagon and other GPCR ligands such as fatty acids and somatostatin. More in general, food intake depends on smell, taste and lipid receptors, while daily metabolic fluctuations are controlled by melanocortin and melatonin receptors [28,29]. The expression of these GPCRs, as well as the different levels of agonists, are likely to be physiologically modulated during childhood and adolescence.

Heterotrimeric G proteins are the canonical GPCR effectors. Gα_s_ and Gα_i_ proteins, respectively, promote or reduce adenylyl cyclase activity and thus cAMP intracellular levels in response to extracellular stimulation. Other G proteins regulate Ca^2+^ signaling and small GTPases activity. From yeast to humans, cAMP remained central to regulate ‘satiety’ linked states, as opposed to promoting appetite and food intake. In yeast, however, the downstream second messenger cAMP might have originally been modulated via G protein-independent pathways. Although poorly characterized, these parallel pathways are still present in higher organisms adding to the complex crosstalk of intracellular signaling mechanisms. In multicellular organisms, GPCRs became pivotal to any aspect of physiology. In humans, GPCR ligands such as incretins regulate glucose homeostasis by increasing insulin secretion from β cells and promoting glucose storage. Vice versa, glucagon produced by α cells reduces insulin secretion to balance glycemia. Glycemic levels need to be continuously adjusted to match energy expenditures throughout the day. Additional GPCR ligands are determinant to this dynamic equilibrium: adrenalin and prostaglandins stimulate blood glucose increase in the condition of imminent external challenges, while cortisol and melatonin are the main players for the physiologic changes in metabolism related to circadian rhythm.

## 3. Type 2 Diabetes and Circadian Rhythm

Circadian rhythms are complex physiological and behavioral systems coordinated by the 24-hour light–dark cycle [30]. Modern lifestyles dramatically modified not only the availability of food but also our exposure to light in terms of spectra and timing. The impact of unnatural exposure to light on glucose metabolism has been studied in rodents [31] and correlations between diabetes and circadian rhythm alterations have been extensively reported in the literature, both for children and adults. Sleeping disorders’ impact on obesity, insulin resistance and type 2 diabetes onset are also well known; however, the underlying molecular mechanisms have not been clarified yet. Melatonin is well known for its role in the regulation of the sleep/wake cycle and other circadian and seasonal rhythms. As a circadian hormone, it peaks during the night in correspondence to natural fasting moment [32,33]. Melatonin pleiotropic action is mediated by two different GPCRs, melatonin receptor 1 (MT1 or MTNR1A) and melatonin receptor 2 (MT2 or MTNR1B) [34,35], known to interact with Gα_i_ and Gα_q_ [36]. Genetic studies confirmed the influence on glucose metabolism of the MT2 gene associating single nucleotide polymorphisms, including partial or complete loss of function variants, with type 2 diabetes [37]. However, it remains to be clarified how melatonin levels affect glucose control and type 2 diabetes risk [38].

Melatonin is naturally produced from serotonin in the pineal gland (PG) and released in the bloodstream [39]. In addition, it is found at high levels in mitochondria where it can activate MT1 receptors. Recent studies reported that melatonin was introduced in eukaryotes by prokaryotes and remained evolutionarily conserved in mitochondria and chloroplasts. In humans, melatonin production is reduced in newborns and then rises between 9 and 12 weeks after birth [33,40]. The highest level of melatonin production is reached between the 4th and the 7th year of age, followed by a continuous decay during the lifespan [41]. Widely administered in children for short-term treatment of sleeping disorders [42], melatonin pharmacokinetics and optimal dosage are still undefined even if it is hypothesized to represent a valid candidate drug for treating type 2 diabetes among different diseases [43]. Melatonin is considered safe; however, doubts have been raised considering its involvement in brain development in children and adolescents [44]. We found no clinical trial regarding melatonin as a drug for type 2 diabetes treatment in children (0–17 years old). Further investigation is required to understand the biology of melatonin signaling in children and, more generally, to dissect the positive vs. negative effects of melatonin on insulin production by β cells. Particularly, it has been suggested that the opposite effects of melatonin on insulin secretion may be related to the fed state; according to the “timing model” proposed by Garaulet et al., melatonin signaling would be strongly biased by feeding and therefore by incretins [38].

Blue light (480 nm) indirectly regulates pineal melatonin production by acting on intrinsically photosensitive retinal ganglion cells (RGCs) sensed by the GPCR melanopsin (OPN4), as reported in Figure 1. Adolescents are exposed to relatively large doses of blue light, the most energetic electromagnetic wave in visible light, generated by smartphones, laptops and tablets, etc., compared with previous generations. Extended exposure to artificial blue light is known to induce retinal degeneration, thus causing mitochondrial damage and ROS burst [45]. It would be interesting to investigate the effects of this exposure on melatonin and insulin circadian fluctuations. Photoresponding opsins (OPN3 and OPN4) present in adipocytes induce lipolysis in response to blue light exposure. In these cells, OPN3 is also crucial for mitochondrial organization and maintenance and regulates glucose and fatty acid transportation. Animal models demonstrated that a high-fat diet down-modulates OPN3 in white adipocytes, and the latter protects from diet-induced obesity and insulin resistance. Exposing brown adipose tissue to led light leads to an increase in glucose uptake and glucose-dependent mitochondrial respiration in the wild type, whilst cells knocked out for OPN3 did not show any consequence after exposure [46]. Modern sleep rhythms and over- or under- exposure to blue light are likely to alter opsin stimulation and in turn to contribute to the insurgency of type 2 diabetes.

Glucocorticoids (GC) represent another class of molecules involved in the circadian rhythm. Among these, cortisol exerts its action by interacting with specific receptors that recently were found to include GPCRs [47]. Cortisol synthesis and secretion are controlled by the hypothalamic–pituitary–adrenal axis with day/night oscillations [48]. Children and adolescents are increasingly stimulated by technological devices, with a reduction in sleeping time and an increased level of stress [49]. Among the different factors which regulate insulin resistance, stress responses play an important role through metabolic and inflammatory signaling [50]. As a consequence of increasing stress levels, cortisol level is reported to increase [51]. The stress response is particularly important in children because of the plasticity of the developing brain with consequent alterations in the nervous and endocrine systems.

Chronically elevated cortisol levels due to dysregulation of the hypothalamic–pituitary–adrenal axis have been associated with increased tissue sensitivity to GC and with chronic activation of GC receptors, thus leading to insulin resistance and glucose intolerance [52]. Moreover, chronically elevated cortisol levels due to Cushing’s syndrome or induced by a two-week exposure to GC raise insulin plasma levels at a basal state, but the organism fails to further increase it in response to a glucose load or standardized meals. This phenomenon is called “relative hypoinsulinemia” and demonstrates the noxious effects of GC on β cells. Nevertheless, the underlying mechanism remains to be clarified [53].

The link between circadian rhythm unbalance and diet-induced obesity is well established. Food availability promotes the stimulation of a circadian oscillator outside of the suprachiasmatic nucleus (SCN) associated with food intake. After eating, fatty acids modulate neuronal activity in the hypothalamus by brain nutrient sensing neurons involved in energy regulation. The desynchronization between the central and the peripheral clocks by altered timing of food intake leads to the disruption of the circadian clock and the increased risk of metabolic disease development [54]. In fact, different studies reported the increased prevalence of type 2 diabetes in night-shift workers [55], highlighting the strong connection between diet and circadian rhythm in relation to type 2 diabetes onset.

## 4. Food Intake and Type 2 Diabetes: Future Pharmacological Strategies for Children

Food ingestion triggers a series of events involving food digestion, nutrient absorption and waste material expulsion. Hormones finely harmonize the fed or hungry state of the organism according to its dynamic energy requirements throughout the day. Several GPCRs are involved and catalyzed attention as anti-obesity targets. Among them, cannabinoid receptor 1 (CB1) [56] and central D2 dopamine receptors are the targets of bromocriptine, a drug that is administered in adults after awakening to act in combination with diet and physical activity. As a result, dopamine circadian rhythms are reset, insulin sensitivity is increased and hepatic glucose production is decreased [57].

Children’s nutrition is highly influenced by the anticipatory and the consummatory food reward, two good feeling responses after respectively seeing or eating highly palatable food, both mediated by the activation of dopamine [58,59]. Anticipatory food reward is usually enhanced by hypercaloric foods such as hamburgers and fries. Both responses are mediated by GLP1 action [60] that reduces the anticipatory food reward and increases the consummatory one, thus reducing appetite and food intake [61,62,63]. This behavioral effect of GLP1 is mediated by vagal afferent neural fibers [64,65,66] and is complemented by many other metabolic effects. The best-known one is the incretinic effect that is the enhanced insulin response in pancreatic β cells induced by oral glucose intake. After sugar consumption, most of the insulin released in the bloodstream depends on the action of GLP1 and glucose insulinotropic peptide (GIP) stimulation [67]. GLP1 relies on bloodstream transportation to stimulate receptors in different organs: in the heart it has a cardioprotective effect stimulating cardiac output, in the stomach it slows down gastric emptying, in the liver it inhibits glucose production, in the muscles it increases insulin sensitivity and in adipose tissue it stimulates lipolysis and reduces inflammation. In the pancreas, GLP1 inhibits glucagon secretion from α cells [68,69].

GLP1 is produced by intestinal enteroendocrine L cells in response to macronutrient digestion products such as monosaccharides, short chain fatty acids, glycerol and medium-long chain fatty acids and amino acids. Except for monosaccharides, which are sensed by the sodium-glucose transporter in the small intestine, all other molecules are sensed by GPCRs. Heterodimeric taste receptors (T1R) recognize all L-type amino acids but tryptophan, calcium sensing receptor (CaSR) recognizes aromatic amino acids in addition to Ca^2+^ and GPRC6A recognizes basic amino acids [70]. GPRC6A also binds to testosterone, Ca^2+^, zinc, magnesium and osteocalcin. The latter was reported to increase β cell proliferation and insulin sensitivity and protect from obesity and metabolic syndrome after mice were fed with a high-fat diet. To date, there is no direct evidence of these effects derived from human studies [71]. Short-chain fatty acids derived from fiber activate free fatty acid receptor (FFAR) 2 and 3, medium-long-chain fatty acids derived from triglycerides activate FFAR1 and 4 [72,73,74,75,76,77]. These receptors also stimulate the release of GIP and peptide YY (PYY).

FFARs have recently been regarded as therapeutical targets for several metabolic alterations such as liver disease, obesity, hyperlipidemia, metabolic syndrome and diabetes. Secor et al. thoroughly described their expression patterns outside the intestine and the relative specific functions [78]. Targeting FFARs with specific agonists could therefore allow treatments that precisely target only the impaired mechanisms of physiologic fed state response in type 2 diabetes patients without altering the whole overall glucose balance [79,80,81,82]. Despite their potential for treating a wide range of morbidities, synthetic FFAR agonists did not reach clinical use, the main reason being a high frequency of adverse reactions [83]. Indeed, important efforts are ongoing to discover agonists that minimize the side effects described above by selectively stabilizing specific receptor conformations and achieving this goal could allow for important developments. There are currently no clinical studies testing FFARs agonists in pediatric type 2 diabetes patients.

Until late 2021, children diagnosed with type 2 diabetes could only benefit from metformin and insulin. After being successfully administered to adults since 2010, the injectable GLP1 agonist liraglutide was tested on children showing efficacy in glycemic control when used together with metformin [84]. This increased efficacy led to FDA approval for liraglutide use in children but came with the cost of increased adverse effects frequency as compared to metformin alone, most of them being gastrointestinal discomfort and nausea [85,86]. Exenatide, another GLP1 agonist which has effects on weight management and only requires weekly injection, was recently approved in 2021 by the FDA for patients between 10 and 17 years old [87]. Adolescents treated with GLP1R agonists were less likely to be prescribed concomitant insulin within one year from diagnosis. Other demographic characteristics or prescriptions of other anti-obesity or anti-diabetes medications did not differ between groups [88]. Several authors emphasized the differences that exist between adults and pediatric patients and the need for clinical trials directed specifically at adolescents [89]. Detailed knowledge of the tissue and subcellular distribution of FFARs and incretins GPCRs may drive the rationale to design novel agonists tailored for pediatric use. The current state of the art for type 2 diabetes pediatric drugs that we present is summarized in Table 1.

## 5. Signaling Integration on the Cell Surface: Biased Agonism, Multi-Agonists and Dimerism

High-affinity interactions of agonists or antagonists stabilize GPCRs in the ‘active’ or ‘inactive’ conformation. Each ligand stabilizes slightly different GPCR conformations, which in turn demonstrate biased efficiency in the interaction with downstream effectors which, in addition to G proteins, include ubiquitous adaptors named arrestins. With an almost complete level of redundancy, two isoforms of arrestins selectively recognize virtually all GPCRs but only in the activated form. The consequent translocation from the cytosol to form a molecular complex leads to GPCR recruitment on coated pits, followed by endocytosis and/or to the activation of G protein-independent responses. FFAR1 was reported to activate anti-inflammatory responses via arrestin, while all the other responses are mediated by Gα_q/11_. Similar indications were obtained for FFAR4 in a recombinant system and in animal models, where arrestin 3 was overexpressed or downmodulated in the pancreas [91,92]. It would be desirable to target FFARs with drugs biased toward arrestin to prevent the loss of islets and other co-morbidities linked to type 2 diabetes-associated inflammation. This aspect appears particularly relevant in childhood and adolescence when the immune system is hyperactive.

Genetic variants of GPCRs can also lead to biased signalling and thus influence the response to treatments; this aspect becomes particularly relevant for tailored treatments and personalized medicine. Direct investigation of 40 type 2 diabetes-associated variants of the MT2 receptor revealed that mutations determining a bias towards arrestin 3 recruitment, rather than Gα_i/o_, have the most statistically significant associations with increased type 2 diabetes risk [93].

A recent review described the implications of biased agonism for therapies targeting GLP1R in type 2 diabetes. The receptor interacts with Gα_s_, Gα_i_, Gα_q_ and arrestin. Different combinations of the effectors produce different outcomes, the main ones being cytoprotective effects, insulin secretion, appetite suppression and nausea. Commercially available GLP1 analogues are differently biased towards arrestin with various internalization rates [94]. Developing specific agonists biased towards G protein signalling might improve GLP1R agonists’ anti-hyperglycemic function and might reduce side effects [95].

Incretins naturally act in combinations and exert a concerted action calibrated on the diet composition [96]. Rather than focusing on single-target therapies, such as insulin, sodium-glucose transport protein 2 (SGLT2) or GLP1, it appears more logical to mimic physiology by the simultaneous intervention of multiple GPCRs agonists such as GLP1, GIP, peptide YY, gastrin, neurotensin, secretin, glucagon, amylin [97]. The achievement of better glycemic control and reduced gastrointestinal side effects would be particularly relevant for pediatric patients that are more exposed to long-term effects and to lower compliance. The dual GIP and GLP1 agonist Tirzepatide is being tested in several phase III clinical trials including the SURPASS-PEDS which is currently enrolling adolescent patients [90]. Preliminary data support its efficacy as the most potent treatment to reduce glucose and body mass in type 2 diabetes. Its effects are not exclusively explained by the reduced appetite [98] but also include an improved response to insulin of β cells [99]. This interesting dual agonist approach could improve specificity for a given cell type and reduce side-effects, by preventing the effects of GLP1 analogues on the CNS and blood pressure, which appear particularly relevant in consideration of chronic treatment.

This strategy brought up high expectations for poly-agonists, regarded as the most promising therapeutic approach to both reducing body weight and improving insulin sensitivity, as recently reported by Iafusco et al. [16]. The combinatorial approach is gaining attention with an additional perspective, i.e., the development of agonists that act simultaneously on two or even three GPCRs. These molecules are often chimeric peptides combining or just fusing the sequences of the natural ligands. These approaches include dual and tri-agonists and have been described in detail elsewhere [96,97]. From this perspective, it appears particularly relevant that GPCRs naturally form dimers and oligomers and that β cells express GPCRs for incretins, serotonin, ATP, etc. Heterodimers of melatonin receptors, MT2 and MT1, or serotonin receptors [100] and heterodimers of GIP and GLP1 receptors have been demonstrated and characterized [101]. Multiple combinations of hetero- and homodimers may thus be formed, each one with its own pharmacology in terms of affinity for the multiple agonists and biases.

## 6. Signaling Integration inside the Cell: GPCR Phosphorylation, Trafficking and Membrane Permeable Agonists

Typically, activated GPCRs stimulate heterotrimeric G proteins and are rapidly phosphorylated by GPCR kinases (GRK) on multiple S/T in their C terminus and the third long intracellular loop of dopamine (D2, D3) and muscarinic (M2) receptors [102,103]. Phosphorylation promotes the interaction with arrestin that in turn mediates the recruitment of a protein machinery instrumental to internalizing the GPCR in endosomes. Similar to cargo transporters, GPCRs can be either directed to lysosomes or unloaded of the ligand and redirected to the plasma membrane to respond to the next round of stimulation. The itinerary can be relatively rapid from early endosomes or can include intracellular compartments, such as the perinuclear recycling compartment [104]. Once internalized, GPCRs are not necessarily silent but can interact with intracellular effectors (G proteins and arrestins) or even with other GPCRs, reshuffling the dimeric composition. The dynamic relocation of GPCRs inside the cell may thus constantly tune the equilibrium and the responsiveness of the cell to signals that cross the plasma membrane or not. A deeper understanding of the equilibria determined by age, sex, diet-related signals could allow a more precise pharmacological intervention for type 2 diabetes treatment. As compared to the natural agonists, dual agonists demonstrated the same efficacy to activate Gα_s_, but reduced and diversified internalization and recycling kinetics to the plasma membrane via the long (rab11 from positive perinuclear compartment) or the short (rab7 positive from late endosomes) pathways. Since GPCRs are also found in cellular organelles, arrestin bias is determinant also to control their subcellular distribution. Indeed, it is also possible that “biased signalling” results from biased trafficking, i.e., that the activation of alternative functions is not due to different direct effectors but to the activation of the same G protein in different subcellular locations.

GPCR internalization does not occur only upon acute stimulation [105]. An in vivo study recently described the important rate of constitutive GLP1R recycling due to spontaneous or paracrine-activated internalization [94]. Constitutive internalization has also been described for FFAR1. In vitro experiments showed how the receptor in the ‘inactive’ conformation was constitutively internalized by a subset of rab5 negative endosomes. Occupied FFAR1 was instead internalized by a distinct population of rab5 positive endosomes and the process required β arrestin 1 and GRK2 [106]. Variations of GPCR ligands due to food or drug intake are thus expected to shift a pre-existing equilibrium, expanding the active intracellular fraction rather than just triggering stimulation over a clean background. Regardless of whether it is due to spontaneous or tonic stimulation, this equilibrium should probably be taken in greater consideration while determining personalized posology. For instance, puberty is associated with increased levels of circulating free fatty acids [107]. In youths, a larger fraction of FFARs can thus be assumed to be normally phosphorylated and recruited into endosomes by arrestins. As a result, the fraction of FFARs present in endosomal compartments and the threshold of their activation by extracellular stimuli are likely increased.

Steroids, hormones and melatonin are among the few natural GPCR ligands that can cross the plasma membrane and diffuse inside cells, including β cells. Estrogen can reach nuclear GPER and melatonin can diffuse to MT1 and MT2 residing on mitochondria. Mitochondrial MT1 signal transduction activates Gα_i_ and blocks adenylyl cyclase activity, leading to the inhibition of stress-induced cytochrome c release and caspase activation. MT1 targeted overexpression in mitochondria inhibited neuronal death resulting from hypoxic/ischemic injury. Recent work has shown how mitochondria synthesize melatonin and directly activate MT1 receptors on their outer membrane [108].

GPCR phosphorylation is an additional important aspect in the control of endosomal sorting and consequent GPCR intracellular localization. Several years ago, it was reported that different phosphorylation domains play different roles in directing the destination of internalized GPCR [109]. Diverse combinations of phosphorylation sites are likely acting as a cipher that encodes the intracellular destination of FFAR [110] or any other GPCR signaling complex. GRK-dependent phosphorylation was defined as homologous, i.e., induced by the activation of the GPCR, as opposed to PKC- and PKA- dependent phosphorylation that contribute to GPCR phosphorylation in response to second messengers activated by the same receptor or others. However, crosstalk mediated by GRK and deviating from the previous definition was observed in response to IGF which was shown to promote FFAR4 phosphorylation, followed by interaction with arrestins and consequent internalization [111]. IGF is primarily produced by the liver in response to GH, and consequently, blood levels peak during puberty. GRKs and arrestins were shown to bind to IGF-1R and to mediate its downstream signaling with distinct GRKs sorting opposing effects, potentially affecting the lifespan of arrestin association. The complex with arrestin 3 favors the recruitment of the E3 ubiquitin ligase Mdm2 to the ligand-free receptor, while the complex with arrestin 2 favors the activation of MAPK by an occupied receptor [112]. Insulin and IGF-1 act on β cells to control their function and are determinant to compensate insulin resistance with an increased proliferation. Insulin and IGF receptors dimerize and are constitutively internalized via clathrin-coated vesicles, establishing an equilibrium described as determinant for a paracrine control of the β cell mass [113]. GRKs and arrestins appear, therefore, as critical signaling knots in the integration of the β cell signaling network.

A detailed description of targetable receptors on β cells while taking into account all the variables described above clearly appears as an overwhelming effort (Figure 2). However, modern techniques of high-content microscopy, single-cell RNAseq, nanobodies that sense protein activation and phosphorylation, in vivo microscopy, all combined with deep learning, will likely provide in the near future the opportunity to analyze these equilibria from a totally different perspective, and possibly lead later to personalized precision medicine.

## 7. Conclusions

From yeasts to mammals, the signaling machinery assembled around GPCR has reached a high level of complexity. The interplay of ligands acting on multiple GPCR structures and multimeric forms converges on multiple effectors in distinct intracellular locations remains. A better understanding of the basis of this architecture could provide the tools to prevent the negative effects of modern children’s lifestyle that cannot be prevented by healthier behavior. Neogenesis, proliferation and apoptosis define the number of β cells that populate the pancreas during life. Multiple endocrine axes coordinate the function of pancreatic endocrine cells, hepatocytes, adipocytes, muscle cells, etc., to adapt glucose metabolism to adult and developing organisms. The expression of over 35 GPCRs has been reported in β cells, highlighting the potential diversity of the four β cell subtypes in terms of function and representation [116]. Being able to model and rationally act on GPCR to regulate this complex equilibrium will provide a new level of precision medicine for pediatric type 2 diabetes patients.

## Figures and Tables

**Figure 1 biomedicines-11-01576-f001:**
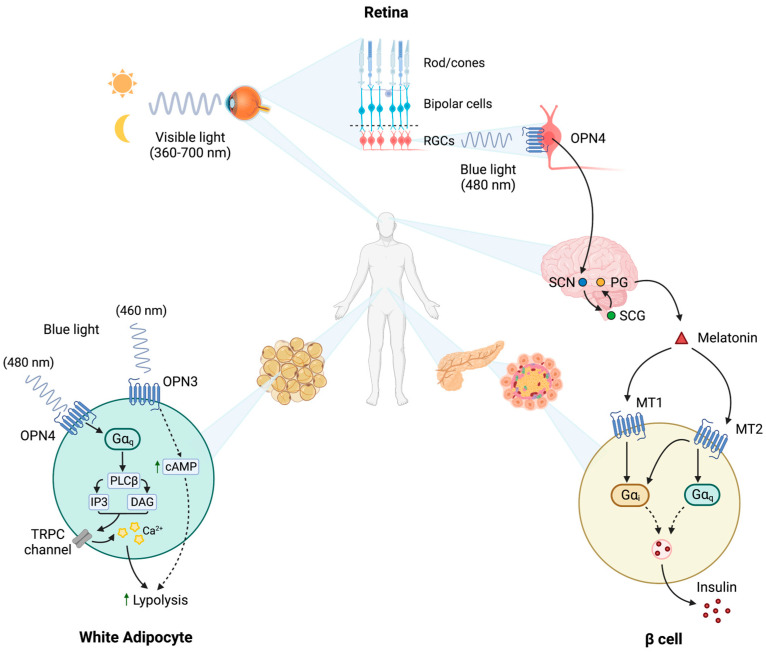
Light perception effects on glucose metabolism—Humans sense visible light primarily through cones and rods in the retina. However, additional opsins specialized to sense blue light are located in the 0.2% of RGCs or in subcutaneous white adipocytes. RGCs project to various areas of the brain to transmit visual information received from the retina. Projections directed to the SCN are instead important to synchronize the circadian clock by regulating melatonin production in the PG through the superior cervical ganglion (SCG). The PG is the most important source of circulating melatonin. Melatonin then regulates insulin release from β cells. 1–5% of the blue light can penetrate the human skin and reach the subcutaneous white adipocytes. Here, OPN4 and OPN3 melanopsin receptors are reported to promote lipolysis. Dotted arrows represent indirect effects of the molecules reported.

**Figure 2 biomedicines-11-01576-f002:**
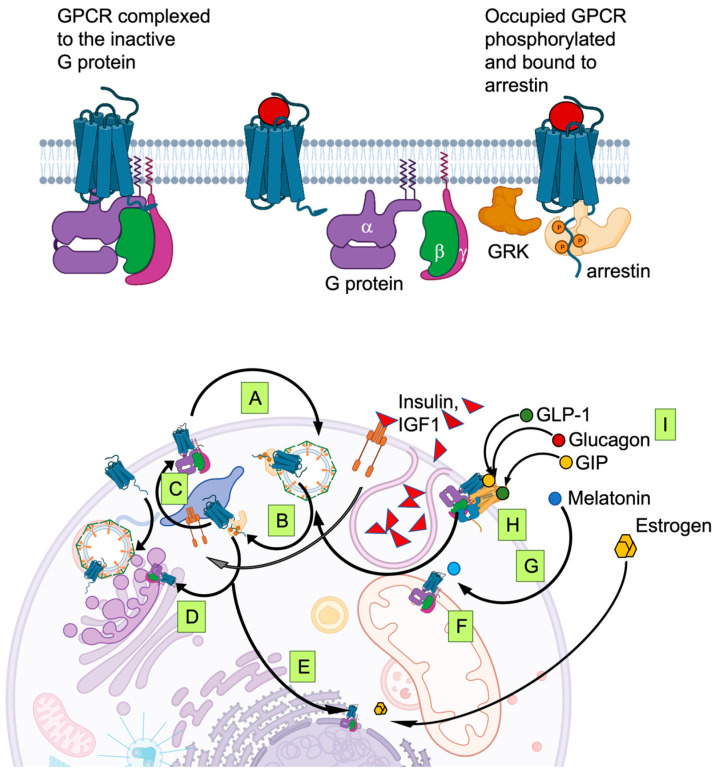
GPCR signaling should be seen as a dynamic equilibrium—Driven by spontaneous activation or driven by the basal concentration of ligand, GPCRs are constitutively internalized (A) and recycled (B). Fluctuations of endogenous ligand will stabilize ‘active’ conformations and thus promote the dissociation of the heterotrimeric G protein and the binding of arrestin to a phosphorylated receptor. Early endosomes can transfer GPCR to different intracellular compartments such as late endosomes (C) or perinuclear recycling compartment (D). The perturbation of the pre-existing balance will redistribute GPCR inside the cell, including nuclear (E) and mitochondrial (F) membranes. Membrane-permeable ligands such as melatonin or estrogens can diffuse inside the cell, directly promoting intracellular signaling (G). GPCR sense stimuli while directly interacting with other GPCRs, as dimers or oligomers, or with other receptors (H) and receptor like GLP1R can recognize multiple ligands, such as glucagon and GLP1 (I). Arrestins and other interactors can sense the activation of distinct receptors. From this description, it appears that coordinated stimuli such as combinations of incretins transmit detailed information about the composition of the meal that will be deciphered by β cells or by neurons in CNS based on the perturbation of an equilibrium that was established during fastening. This complex machinery evolved from prokaryotes will allow a coordinated response of the organism in terms of satiety, enzyme production, etc. Fasting glucose, insulin and insulin resistance change during childhood [114] and incretin levels may be titrated to favor growth during adolescence [115]. The design of pharmacological treatments should consider this level of complexity to achieve the best efficacy and minimize undesired effects during adolescence and puberty.

**Table 1 biomedicines-11-01576-t001:** Drugs for type 2 diabetes in pediatric patients—the table summarizes the drugs for young type 2 diabetes patients which are currently available or under research to date. Abbreviations: glucagon like peptide 1 receptor (GLP1R), glucagon like peptide 1 (GLP1), melatonin receptor 1 and 2 (MT1 and MT2), photoresponding opsins 3 and 4 (OPN3 and OPN4), G protein-coupled receptor class C member A (GPRC6A), free fatty acids receptors (FFARs), gastric inhibitory polypeptide (GIP), peptide YY (PYY), gastric inhibitory polypeptide receptor (GIPR).

GPCR	Ligand	Effects	Research Phase	Reference
GLP1R	GLP1	Insulin secretion and glucose uptake increase, appetite reduction, gastric emptying inhibition	Commercialized drugs (liraglutide, exenatide, semaglutide)	[21]
MT1 and MT2	Melatonin	Circadian rhythm regulation, appetite modulation	Pre-clinical	[43]
OPN3 and OPN4	Light	Diet-induced obesity and insulin resistance protection	Pre-clinical	[46]
Cortisol G protein-coupled receptor	Cortisol	Insulin resistance and glucose intolerance after chronic stimulation	Pre-clinical	[52]
GPRC6A	Testosterone, Ca^2+^, zinc, magnesium, osteocalcin	β cell proliferation and insulin sensitivity increase, obesity and metabolic syndrome protection	Pre-clinical	[71]
FFARs	Short-, medium- and long-chain free fatty acids	GLP1, GIP and PYY release stimulation	Pre-clinical	[78]
GIPR-GLP1R	GIP, GLP1	Incretinic effect stimulation	Phase III clinical trials (Tirzepatide)	[90]

## Data Availability

Not applicable.

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
