# Peer review of "G Protein-Coupled Receptors and the Rise of Type 2 Diabetes in Children"

_biomedicines, 2023, doi:10.3390/biomedicines11061576_

Round 1

Reviewer 1 Report

This manuscript by Alessia Dallatana et al., presents a “G protein coupled receptors and the rise of Type 2 Diabetes in children” Review for GPCR expression functional relation in the young generation with type 2 diabetes. 

Here is I have minor concerns.

Instead “T2D” should be using “type 2 diabetes” for the reader. 

Line 54: GPER1 is G protein-coupled estrogen receptor, in general, “estrogen receptor means ESR1 or ER alpha. Please clarified it. Also, amino acid sensing GPCR is not only GPRC-6A only, so add more GPCR or remove the GPRC-6A word. In addition, authors may need references for those …

Line 172: (480nm) change to (480 nm)

Line 220: 0,2% change to 0.2%

Line 263: for FFAR need add “free fatty acid receptor.”

Line 325: what is SGLT2?

Line 347: the authors need to explain what are MT1 and MT2.

Line 355: GRK phosphorylates GPCR not only c-terminal but also IC3 loop also, so authors need to address this. 

Line 356: the authors may switch to “induce the interaction” rather than “stabilize the interaction.’

Figure 2: in membrane push back and GPCR bring to front may better to see. inside g proteins need to describe with alpha, beta, gamma also GRK, and beta-Arrestin.

Author Response

We thank the Reviewer 1 for his very helpful feedback. We reviewed the manuscript and modified it as suggested.

- Instead “T2D” should be using “type 2 diabetes” for the reader.

The acronym was reverted to the full name.

- Line 54: GPER1 is G protein-coupled estrogen receptor, in general, “estrogen receptor means ESR1 or ER alpha. Please clarified it. Also, amino acid sensing GPCR is not only GPRC-6A only, so add more GPCR or remove the GPRC-6A word. In addition, authors may need references for those …

We modified the sentence and added the references where required.

- Line 172: (480nm) change to (480 nm)

We changed (480nm) to (480 nm).

- Line 220: 0,2% change to 0.2%

We changed 0,2% to 0.2%.

- Line 263: for FFAR need add “free fatty acid receptor.”

We defined FFAR.

- Line 325: what is SGLT2?

We defined SGLT2 (sodium-glucose transporter protein 2).

- Line 347: the authors need to explain what are MT1 and MT2.

We added an explanation for MT1 and MT2.

- Line 355: GRK phosphorylates GPCR not only c-terminal but also IC3 loop also, so authors need to address this.

We modified the sentence and we added a reference.

- Line 356: the authors may switch to “induce the interaction” rather than “stabilize the interaction.’

            Since phosphorylation is not a requirement for arrestin interaction (see references below), we preferred the term “stabilize”. Arrestin senses the conformational change in the GPCR, which is paralleled by GRK activity. The combination of phosphorylation sites has different consequences in directing the interaction with arrestin and the functional effects remain to be fully understood. In order to keep into account the reviewer suggestion, we used a more conservative verb (i.e. “promote”) to cover all concerns.

Chen, H., Zhang, S., Zhang, X. et al. QR code model: a new possibility for GPCR phosphorylation recognition. Cell Commun Signal 20, 23 (2022). https://doi.org/10.1186/s12964-022-00832-4

Seyedabadi M., Gharghabi M, Gurevich EV and Gurevich. Receptor-Arrestin Interactions: The GPCR Perspective

Biomolecules 2021, 11(2), 218; https://doi.org/10.3390/biom11020218

- Figure 2: in membrane push back and GPCR bring to front may better to see. inside g proteins need to describe with alpha, beta, gamma also GRK, and beta-Arrestin.

The figure was modified as suggested.

Reviewer 2 Report

The authors have reviewed GPCRs relevant for b cells function with an emphasis on aspects that could help to differentiate new treatments for young T2D patients. I have the following concerns regarding the manuscript:

i.                     Under the heading ‘T2D and circadian rhythm’, authors should define circadian rhythm first.

ii.                   The heading ‘Signalling integration inside the cell: GPCR phosphorylation, trafficking and membrane permeable agonists’ Is not necessary to put. If the authors want to connect the signaling mechanism to the T2D, or β-cells, they can mention it briefly in a few sentences. The whole signaling mechanism may not be necessary as the main focus of the manuscript is on T2D in children rather than the signaling mechanism.

iii.                 The manuscript is long. There should be a table that shows the GPCRs that may be targeted in T2D in children. Likewise, the GPCRs that have been used as pharmacological targets in T2D in children must also be listed in the form of a table.

iv.                 The abstract can be improved.

v.                   There are some spelling errors.

Some grammatical and spelling errors should be revised.

Author Response

We thank the Reviewer 2 for his very helpful feedback. We reviewed the manuscript and modified it as suggested.

i. Under the heading ‘T2D and circadian rhythm’, authors should define circadian rhythm first.

We inserted the definition of circadian rhythm at the beginning of the paragraph entitled “Type 2 diabetes and circadian rhythm”.

ii. The heading ‘Signalling integration inside the cell: GPCR phosphorylation, trafficking and membrane permeable agonists’ Is not necessary to put. If the authors want to connect the signaling mechanism to the T2D, or β-cells, they can mention it briefly in a few sentences. The whole signaling mechanism may not be necessary as the main focus of the manuscript is on T2D in children rather than the signaling mechanism.

This paragraph and the previous one are meant to describe general mechanisms of GPCR signalling that are relevant to T2D. The second paragraph depicts the recent vision of intracellular signalling that we believe is particularly important to develop specific drugs for young patients. This is because age is associated with variations of the levels of stimuli rather than intrinsic characteristics of the receptor, such as genetic background, ligand affinity etc. These variations imply a different distribution of the GPCRs and effector molecules in the cells.

We added a sentence to address this point and we slightly simplified the beginning of the previous paragraph that introduced the signalling mechanism. We believe that a brief description of dual agonists and intracellular trafficking was necessary to introduce the second part of the paragraph dealing with FFARs, MT1/2 and IGF1-R, kinases and arrestin. The paragraph dedicated to the description of how single signals are processed and integrated over time and inside the cells, was broken in two mainly because of its length. We realize it is relatively long, however, we could not shorten it since the goal was to describe the complexity of the system in perspective of designing more precise therapies.

iii. The manuscript is long. There should be a table that shows the GPCRs that may be targeted in T2D in children. Likewise, the GPCRs that have been used as pharmacological targets in T2D in children must also be listed in the form of a table.

We are grateful to the reviewer for this advice. A table including both your suggestions was added to the manuscript.

iv. The abstract can be improved.

We modified and improved the abstract as suggested.

v. There are some spelling errors.

We checked and modified the spelling errors.

Reviewer 3 Report

Based on the title, I had expected a more clinical review rather than a discussion of potential GPCR targets for treating type II diabetes. I detected the following minor issues:

References 9, 33, 48, 59, 66 and 114 are incomplete.

References 93 and 104 are identical.

Some of the DOIs in the references link to single files (e.g. a particular supplementary figure) rather than the whole referenced paper.

The sentence in lines 255-257 does not make grammatical sense.

Author Response

We thank the Reviewer 3 for his very helpful feedback. We reviewed the manuscript and modified it as suggested. 

- References 9, 33, 48, 59, 66 and 114 are incomplete.

We modified references 9, 33, 48, 59, 66 and 114 as suggested.

- References 93 and 104 are identical.

We corrected the references 93 and 104.

- Some of the DOIs in the references link to single files (e.g. a particular supplementary figure) rather than the whole referenced paper.

We corrected the references as suggested.

- The sentence in lines 255-257 does not make grammatical sense.

We modified the sentence in line 255-257 as suggested.